# Platelet–Neutrophil Crosstalk in Thrombosis

**DOI:** 10.3390/ijms24021266

**Published:** 2023-01-09

**Authors:** Laura J. Mereweather, Adela Constantinescu-Bercu, James T. B. Crawley, Isabelle I. Salles-Crawley

**Affiliations:** 1Centre for Haematology, Department of Immunology and Inflammation, Imperial College London, London W12 0NN, UK; 2Vascular Biology Research Centre, Molecular and Clinical Sciences Research Institute, St. George’s University of London, London SW17 0RE, UK

**Keywords:** platelets, neutrophils, arterial thrombosis, deep vein thrombosis, neutrophil extracellular traps

## Abstract

Platelets are essential for the formation of a haemostatic plug to prevent bleeding, while neutrophils are the guardians of our immune defences against invading pathogens. The interplay between platelets and innate immunity, and subsequent triggering of the activation of coagulation is part of the host system to prevent systemic spread of pathogen in the blood stream. Aberrant immunothrombosis and excessive inflammation can however, contribute to the thrombotic burden observed in many cardiovascular diseases. In this review, we highlight how platelets and neutrophils interact with each other and how their crosstalk is central to both arterial and venous thrombosis and in COVID-19. While targeting platelets and coagulation enables efficient antithrombotic treatments, they are often accompanied with a bleeding risk. We also discuss how novel approaches to reduce platelet-mediated recruitment of neutrophils could represent promising therapies to treat thrombosis without affecting haemostasis.

## 1. Introduction

Platelets and neutrophils are abundant cells in peripheral blood. Classically, platelets exert an essential haemostatic role, while neutrophils are important for innate immune responses. The essential role of platelets in haemostasis is consistent with the bleeding diathesis associated with thrombocytopenia and platelet function disorders [1,2]. Similarly, neutropenia and loss of neutrophil function have been linked to severe, life-threatening bacterial infections [3,4]. Beside their classical roles, the involvement of platelets and neutrophils in thrombosis and inflammation has received considerable attention within the past decade. It is clear that, unwanted/excessive platelet–neutrophil interactions can occur in certain inflammatory conditions, such as atherothrombosis, atherosclerosis, stroke, deep vein thrombosis (DVT), diabetes, heart failure, sepsis, transfusion-related acute lung injury (TRALI), inflammatory bowel disease [5,6,7,8,9,10,11,12,13]. In 2013, Engelmann and Massberg formulated the term immunothrombosis to describe the physiological process by which the coagulation pathway is activated to limit the spread of a pathogen within the bloodstream [14]. The main drivers of immunothrombosis are platelets and innate immune cells including neutrophils, monocytes and macrophages. Whereas immunothrombosis is beneficial to prevent the invading pathogens entering the bloodstream, in certain inflammatory conditions such as atherosclerosis or DVT, it can lead to collateral damage of organs by excessive thrombus formation. In this review, we describe the various receptors and ligands involved in platelet–neutrophil interactions and highlight how mutual activation of platelets and neutrophils can influence various inflammatory pathological conditions, focusing on thrombosis. Finally, current and future approaches to target thrombosis where the formation of the thrombus constitutes a fatal endpoint will also be discussed.

## 2. Platelet–Neutrophil Complexes Drive Thrombosis and Inflammation

Haemostasis is a highly regulated physiological process that has evolved to maintain the integrity of the cardiovascular system and prevent blood loss. It is defined by two distinct stages, primary and secondary haemostasis. Primary haemostasis is comprised of the accumulation, activation, and aggregation of platelets at the site of vessel injury, whereas secondary haemostasis refers to the activation of an enzymatic cascade, resulting in deposition of fibrin. Despite these being distinct processes, they occur simultaneously and are mechanistically interdependent. Captured platelets at the site of injury are exposed and respond to a plethora of agonists (e.g., VWF, collagen, thrombin, ADP, thromboxane, fibrin) and, as a result, become activated depending on the agonist(s) and their concentration [15,16]. Platelet response is therefore “tuneable” to ensure the formation of the haemostatic plug. Haemostasis is tightly regulated, but under pathological conditions can lead to arterial, venous and microvascular thrombosis. Classical understanding of the differences between these historical disorders suggests that arterial thrombi appear at regions of vessel injury, most commonly following rupture of atheromas, where sub-endothelial matrix proteins such as collagen are exposed to components of blood that can initiate thrombosis. This process occurs at high shear rates where platelets are quickly recruited to the site of injury. Thrombi that form are therefore described as predominantly platelet-rich or ‘white clots’. In comparison, venous thrombi form on top of an intact endothelial layer over longer periods of time and at lower shear rates. These are classically described as fibrin-rich or ‘red clots’ and can be heterogenous with areas of both red and white thrombi. As such, current therapy consists primarily of anti-platelet agents for arterial thrombosis and anticoagulant treatment for venous thrombosis. However, the mechanisms behind the formation of both arterial and venous thrombi are complex and involve inflammatory processes that result in more intricate mechanisms that drive thrombosis.

Platelet–neutrophil interactions are known to be important for the normal haemostatic response, refs. [17,18] in innate immunity to facilitate bacterial infection clearance, refs. [19,20] and perhaps more importantly to pathological thrombus formation [6,21,22]. In circulation, platelets and neutrophils do not readily interact. Therefore, known platelet–neutrophil interactions require the platelet and/or the neutrophil to be activated or primed to promote cell–cell binding (Figure 1). For example, activated platelets undergo degranulation and through this process present important leukocyte receptors on their surface, e.g., P-selectin and CD40 ligand (CD40L) that will interact with their respective receptors constitutively expressed on the surface of neutrophils, P-selectin glycoligand-1 (PSGL-1) and CD40 [9,23,24,25,26,27]. Engagement of P-selectin with PSGL-1 leads to activation of integrins (Mac-1 and LFA-1 also termed CD11b/CD18/α_M_β_2_ and α_L_β_2,_ respectively) via downstream signalling of tyrosine kinases [28,29]. Once activated, Mac-1 stabilises platelet–neutrophil interactions by binding directly to platelets through GPIbα, as well as JAM-3 and ICAM-2, or indirectly to activated α_IIb_β_3_ via a fibrinogen bridge (Figure 1) [30,31,32,33,34]. Activated platelets also facilitate neutrophil recruitment and activation through the release of soluble mediators from granules including chemokines (CCL5/RANTES, CXCL4/PF4, CXCL5, CXCL7), serotonin, but also the damage-associated molecular pattern (DAMP) high motility group protein B1 (HMGB1) [35,36,37,38]. Vice-versa, neutrophil activation leads to granule release including cathelicidin (LL-37), myeloperoxidase (MPO), production of reactive oxygen species (ROS), and release of neutrophil extracellular traps (NETs) all promoting platelet activation and thrombus formation [22,39,40]. It is clear that platelet–neutrophil interactions lead to reciprocal activation of either cell type and can influence thrombus formation (Figure 1).

Alongside platelets, neutrophils play a central role in venous and arterial thrombosis where they have been reported to be recruited early and to represent the most abundant leukocytes at the thrombus site [10,41]. Neutrophils can contribute to the initiation of thrombosis through a number of mechanisms, in particular with the release of highly thrombotic NETs [10,22,40]. Both animal and human studies have demonstrated the presence of NETs within a variety of thrombi, refs. [10,42,43,44,45] although in many cases it is difficult to ascertain whether their formation occurred during or post thrombus formation. Evidence suggests that NETs may be an important driver of thrombus development favouring red blood cell entrapment, platelet aggregation and activation of the coagulation cascade. NETs can bind VWF and fibrinogen to which platelets can bind and become activated [46,47]. Platelet activation could also be mediated by histones, as histones H3 and H4 have been shown to trigger platelet activation and aggregation both directly via toll-like receptors (TLR) 2 and 4, and indirectly via fibrinogen [45,48]. Moreover, NETs are negatively charged and, as such, can bind and activate FXII, initiating the intrinsic coagulation cascade [10]. The importance of the contact pathway activation in thrombosis is exemplified by several promising anticoagulants targeting FXI (downstream of FXIIa) currently in Phase II clinical trials [49]. Finally, NETs may also modulate anticoagulant pathways via histones targeting thrombomodulin (TM)-mediated activation of protein C or via their neutrophilic granular content [50,51]. This includes neutrophil elastase (NE), which was associated with the ability to inactivate tissue factor pathway inhibitor (TFPI) [50,51]. 

NET release has been considered to be a host innate immune response to entrap and kill pathogens entering the bloodstream. Locally, initiating haemostatic mechanisms may serve to prevent systemic spread. However, if these mechanisms are aberrantly or excessively activated, these same protective processes can augment thrombus formation. For this reason, understanding the mechanisms underlying platelet–neutrophil interactions and what drives NET production are of particular interest.

## 3. Arterial Thrombosis

Arterial thrombi generally form following an atherosclerotic plaque rupture, through an inflammatory process known as atherothrombosis. Platelets play a particular important role both in the initiation and progression of atherogenesis and in the aftermath of the plaque rupture [52,53]. This is exemplified by the success of anti-platelet therapies in the treatment and prevention of myocardial infarction (MI) and stroke [54]. The rupture of an atherosclerotic plaque leads to the exposure among other things of collagen and TF that trigger platelet adhesion and potent platelet activation together with coagulation initiation [55,56]. Activated platelets release secondary mediators including ADP, TxA2, leading to further platelet activation and aggregation. This ultimately, but not always, leads to a vaso-occlusive thrombi mostly formed of platelets that promote the deposition of large amounts of fibrin (Figure 2) [57].

Monocytes play an important role in atherogenesis where platelets facilitate their adhesion and extravasation through the endothelium [53,56]. The role of neutrophils and platelet–neutrophil interactions in this process and acute coronary events have not been studied as extensively as for monocytes, but recent studies suggest they may facilitate some of these processes.

First, neutrophil counts were found to be directly correlated to MI size and left ventricular function in patients undergoing percutaneous coronary intervention (PCI) for acute ST-segment elevation MI (STEMI) [58]. The large clinical study CALIBER also suggests that the neutrophil count was associated with the incidence of some cardiovascular diseases including MI [59]. Secondly, neutrophils and NETs were found in human carotid atherosclerotic lesions and are thought to be associated with a plaque instability phenotype [60,61]. A recent study using mouse models of atherosclerosis, suggests that activated smooth muscle cells (SMC) recruit and activate neutrophils in the plaque to form NETs and that histone H4 lyses SMC destabilizing the plaque [62]. Inhibition of Peptidyl arginine deiminase 4 (PAD4), an enzyme required for NET formation, reduced plaque burden and thrombosis [61,63,64]. Thirdly, NETs were identified in human coronary thrombi from patients undergoing PCI as well as in thrombi retrieved from mouse arterial thrombi [65,66,67,68]. Disrupting NET formation by blocking PAD4 function or by DNase prevented endothelial injury and reduced thrombus formation [61,64,65]. Finally, the crosstalk between neutrophils and platelets in arterial thrombosis has been highlighted by several studies, mostly in mouse models. Neutrophil derived cathelicidin LL-37 was highly abundant in thrombi from acute MI patients and its mouse homologue CRAMP was shown to activate the GPVI-signalling pathway and facilitate further platelet aggregation [39]. Next, platelet derived HMGB1 activated both platelets via the TLR4-Myd88 pathway and neutrophils via RAGE promoting NET formation [68,69]. The inhibition of platelet-HMGB1 reduced neutrophil infiltrates and NETs, providing protection against thrombosis in an arterial thrombosis model [69]. An elegant study analysing the platelet proteome from MI patients, showed that elevated levels of the neutrophil-derived DAMP S100A8/9 found in patient plasma influenced platelet reactivity [70]. Additional mechanisms of interaction between neutrophils and platelets may involve the P-Selectin/PSGL1 axis or GPIbα/MAC-1 and will be discussed in “Targeting platelet–neutrophil interaction for thrombosis treatment” section.

Together with MI, one of the most serious clinical outcomes that can arise as a consequence of atherothrombosis is stroke. Stroke is a leading cause of cardiovascular morbidity and mortality worldwide. About 80% of strokes are caused by cerebral ischaemia, resulting from a thrombus occluding cerebral blood vessels. Acute treatments of stroke involve thrombus removal by thrombectomy, or thrombolysis with tissue plasminogen activator (tPA). However, even after successful reperfusion the infarct sizes can increase due to inflammatory responses (e.g., inflammatory cytokines release and ROS production) [8]. Therefore stroke is now defined as a thrombo-inflammatory condition, as a link has been found between the initial thrombotic cause and the downstream inflammatory response in the severity of infarct size and ischaemia-reperfusion [8]. Although platelet involvement in stroke has previously been well characterised as part of the underlying thrombotic cause, it recently became apparent that platelets also play a crucial role in the subsequent stroke-related inflammatory pathology. This was shown to be linked to the ability of platelets to release pro-inflammatory chemokines, such as interleukin 1α (IL-1α), [71] but also to platelet capacity to bind endothelial cells and help in the recruitment of neutrophils and monocytes to the stroke site. The inflammatory involvement of platelets in stroke is thought to be mediated via GPVI, as well as the VWF A1-GPIbα axis, but not α_IIb_β_3_ [72,73,74]. In line with this, recent clinical trials with GPVI antagonists showed promising outcomes in the treatment of acute ischemic stroke [75,76,77]. Neutrophils were detected within hours of stroke onset and their numbers correlated with the infarct size in humans and mice [78,79,80]. Another recent study revealed the importance of neutrophils in downstream microvascular thrombosis post middle cerebral artery occlusion in rats [81]. The involvement of neutrophils in ischaemic stroke has gained further support from studies analysing thrombi from ischaemic stroke patients revealing that NETs were an important component of the thrombi [66,67,82,83]. Interestingly, platelet-rich areas in stroke thrombi were accompanied with fibrin rich-structures embedded with VWF, leukocytes and DNA in and around these platelet-rich regions [84]. The importance of neutrophil–platelet interactions in stroke was further demonstrated by several human studies where neutrophil–platelet aggregates were associated with a higher risk of stroke or worse stroke outcome [79,85,86]. Lastly, mouse models of stroke certainly have unequivocally demonstrated the importance of platelet-mediated NET formation via release of HMGB1, and their pro-coagulant potential [78,87].

Although there is ample evidence that neutrophils are present in arterial thrombi and through NETs participate in the formation of an occlusive thrombus perhaps their reported stabilising effect on clots/resistance to thrombolysis may be of greater significance [77,88,89,90]. A recent study suggests that NETs could play an important role in the peri-infarct region during neovascularisation and that stroke recovery could be improved by inhibiting NET formation with DNase or blocking PAD4 [91]. Improved thrombolytic therapy by using DNase treatment has been observed in several studies, but there is yet a clinical trial to target NETs for the treatment of cerebral and coronary thrombolysis [41,77,92,93]. Interestingly, Carminita et al. reported antithrombotic properties of DNase on platelet and fibrin formation in a laser-induced thrombosis model in mice as a result of the cleavage of ADP/ATP into adenosine by DNase, inhibiting not only neutrophil functions but also platelets [94].

## 4. Deep Vein Thrombosis

Venous thromboembolism (VTE), which comprises deep vein thrombosis (DVT) and pulmonary embolism (PE), is a leading cause of cardiovascular morbidity and mortality worldwide and is identified as an inflammatory disorder [95]. The yearly costs associated with the treatment of VTE-related incidents were estimated to be $7–10 billion in the US in 2016 and £640 million in the UK in 2004, representing a great economic burden to their respective healthcare systems, which has likely risen in recent years [96,97]. Compared to other cardiovascular diseases, the incidence of DVT and its associated complications, such as PE, continues to increase [98]. In both the UK and the US, DVT occurs in 1/1000 people per year and this rises to 1/100 in people over the age of 50 years old [98]. Numerous risk factors have been identified including genetic predisposition, obesity, smoking, sedentarism, long-term immobility, and surgery/trauma [99]. Virchow described, over 150 years ago, three factors contributing to thrombosis: blood hypercoagulability, stasis and vessel wall irritability that still stand today [100].

DVT develops in the venous system and in the absence of overt vessel damage, most frequently around venous valves (Figure 3) [101,102,103]. Alterations to flow patterns within venous valves pockets has been associated with DVT risk factors such as immobility or paralysis [102,103,104]. In this instance, within the deep recesses of the pocket, there are regions of disturbed flow and most notably, stasis [101]. Hence, this alteration to flow patterns within valve pockets has been hypothesised as a driver of DVT. As the endothelium can sense shear stress, EC phenotype is influenced by such changes to blood flow patterns, altering levels of anticoagulant proteins and leukocyte adhesion molecules on the cell surface [105]. Studies have highlighted that ECs within healthy valve pockets have a more antithrombotic phenotype than those lining the vein lumen. This includes higher levels of TM, endothelial protein C receptor (EPCR) and TFPI, alongside lower levels of VWF, ICAM-1 and P-selectin [104]. However, this phenotype was lost upon flow-restriction in mice as well as at sites of primary DVT in humans [104]. Therefore, the switch from an anticoagulant to a prothrombotic endothelial phenotype is associated with disturbed flow and initiation of thrombosis. Disturbances in venous flow, particularly stasis, are also associated with hypoxia and can alter levels of plasma proteins and blood cells due to lack of influx or efflux. Although the mechanisms associated with thrombus initiation are not fully understood, endothelial activation is believed to be an early event and contributes to the recruitment of platelets and leukocytes that participate together with red blood cells and coagulation factors to the initiation and growth of the thrombus [106].

Animal models of thrombosis have shed light on the involvement of platelets and neutrophils in venous thrombosis. In a mouse stenosis thrombosis model, neutrophils were observed as early as 1 h after inferior vena cava (IVC) restriction and observed to carpet the endothelium within 5–6 h [10]. The analysis of the subsets of leukocytes accumulating in the venous thrombus in the IVC stenosis model revealed that the majority (~70%) of these were neutrophils. A small proportion of leukocytes were monocytes (~30%), ref. [10] although the timeline of their recruitment in relation to neutrophils has not been determined. TF present on circulating monocytes and TF-positive microvesicles are thought to potentially contribute to thrombus development. This provides an important source of intravascular TF that can initiate the extrinsic coagulation cascade (Figure 3) [101].

Additionally, neutrophils significantly contribute to the development of DVT through their ability to release highly thrombotic NETs [44,47,107]. During VTE and for at least 1 year post event, patients exhibit increased levels of activated neutrophils and circulating MPO-DNA complexes [108]. In mice, depleting neutrophils, administering DNase for the dissolution of NETs or inhibiting the process of NET formation by knocking out *Pad4*, have all been associated with DVT protection [10,44,107,109]. Taken together, such studies suggest that NETs have numerous roles in thrombus propagation. Histological analysis of human venous thrombi indicated NETs are predominantly detected in thrombi in the organisation phase compared with mature thrombi, suggesting they may be important for the early stage of thrombus formation [43]. As previously mentioned, they are involved in thrombus growth by entrapping red blood cells and binding additional platelets. They can activate endothelial cells through the release of proteases (e.g., cathepsin G), increase further leukocyte recruitment to the vessel wall, [110,111] and propagate inflammasome activation, enhancing monocyte pyroptosis [112,113]. NETs can also initiate the intrinsic coagulation cascade through activation of coagulation factor XII, promoting fibrin formation (Figure 3) [10,44,47,107]. FXI-knockout mice showed reduced thrombotic burden in the ferric chloride-induced vena cava model [114]. Similarly, mice deficient in FXII also displayed reduced thrombus formation in IVC restriction models although, interestingly, no thromboprotective effects were observed with FXI deficiency in this study [10]. This also highlights the variability between different models of DVT and different mechanisms underlying thrombus formation. Finally, several studies demonstrated a role for neutrophils/NETs in modulating the anticoagulant function of the endothelium via histones or NE [50,51]. The importance of NE in NET and thrombus formation via modulation of natural anticoagulants remain unclear in light of animal studies showing no protection of NE deficient mice in DVT [115]. Similar to their role in arterial thrombosis, it is still unclear whether NETs are more important for the initiation or propagation of DVT or if they represent a key factor to thrombus stability and resistance to fibrinolysis. They may, in fact, be important in for all these different thrombogenic mechanisms.

The data implicating the importance of platelet-mediated recruitment of leukocytes, especially neutrophils, in the setting of DVT has been particularly compelling, at least in mouse models. Similar to neutropenia, platelet depletion in mice is protective in the mouse stenosis model and importantly also diminishes leukocyte numbers, suggesting that platelets are of major importance in the recruitment of leukocytes in this DVT model [10]. There is also a large body of evidence highlighting the specific importance of the VWF A1-GPIbα interaction in DVT. Numerous studies have reported that blockade of this interaction is thrombo-protective. *Vwf*^−/−^ mice have reduced platelet adhesion to venous thrombi, and smaller sized clots in vivo [42,116]. In addition, pharmacological inhibition with small molecules such as snake venom-derived antibatide, or antibodies against GPIbα are protective against thrombus formation [42,117,118]. Furthermore, mice lacking the extracellular domain of GPIbα on platelets were protected from venous thrombosis upon flow-restriction of the IVC [10]. Importantly, disrupting the VWF A1-GPIbα interaction greatly diminished leukocyte/neutrophil recruitment in the stenosis model [10,42]. Additional studies have highlighted the important role of platelet releasates (e.g., HMBG1) in leukocyte recruitment, activation and NET formation for DVT propagation [38,119]. Finally, within venules, neutrophils have been shown to influence the local rheology, promoting platelet aggregation and platelet–neutrophil interactions, subsequently facilitating thrombus formation [120].

## 5. Novel Concepts in DVT Initiation—VWF-Primed Platelets Recruit Neutrophils

Given recent evidence suggesting that *Vwf*-deficient mice, ref. [42] as well as mice lacking the extracellular domain of GPIbα exhibited diminished leukocyte recruitment following stenosis of IVC and were protected against DVT, ref. [10] it can be inferred that an appropriate proportion of leukocyte capture at sites of DVT occurs in a VWF- and platelet-dependent manner. Previous studies have highlighted a role for P-Selectin and HMBG1 released from activated platelets in NETosis during the formation of a venous thrombus [10,38,119,121,122]. However, the initial interactions taking place in DVT between platelets and neutrophils/leukocytes on the endothelium are not fully understood as, given the lack of endothelial denudation, platelets are not exposed to potent agonists (i.e., collagen, thrombin) that classically drive robust platelet activation and initiate secondary messenger release. Therefore, stimuli capable of inducing P-selectin exposure on their surface appear not to be significant in the early phases of DVT. In line with this, studies show that whereas endothelial P-selectin in part contributes to the development of DVT, platelet P-selectin plays a non-significant role [10]. On the other hand, platelet depletion or disruption of the VWF–GPIbα axis also protect against DVT and prevent recruitment of leukocytes to the endothelium (and therefore subsequent NET formation) implying that VWF-dependent platelet recruitment is necessary for the initiation of DVT. Together, these data also potentially make it unlikely that endothelial P-selectin exposure is the primary event that leads to leukocyte recruitment. As neutrophils as well as most other leukocytes express PSGL-1 (e.g., lymphocytes, monocytes) [24], this mechanism is not selective for a particular leukocyte subset. Moreover, neutrophils appear to represent the major leukocyte population during the early stages of DVT, suggesting a more specific mode of recruitment [10]. Considering that all the previously characterised platelet–leukocyte interactions generally require the platelets to be activated, the selective interaction between platelets and leukocytes responsible for the early recruitment warrants careful consideration.

We recently discovered a novel interaction between platelets and neutrophils, which may be key to the initiation of DVT (Figure 3). Our results show that platelets captured by VWF can become ‘primed’ under flow. This involves activation of α_IIb_β_3_ on the platelet surface but does not induce appreciable degranulation/P-selectin exposure. These ‘primed’ platelets were able to recruit neutrophils under venous shear rates, with more interactions occurring in regions of disturbed flow. The receptors involved in this novel interaction are activated α_IIb_β_3_ on platelets and SLC44A2 on neutrophils [123]. Importantly, neutrophils interacting with VWF-‘primed’ platelets undergo phenotypic changes, including Ca^2+^ release and culminating with the formation of NETs [123]. We also demonstrated the importance of shear forces for the trigger in this process, as neutrophils captured on activated α_IIb_β_3_ under static conditions were unable to form NET [123].

Also known as choline transporter-like protein-2 (CTL-2) or human neutrophil antigen-3 (HNA-3), SLC44A2 is a transmembrane protein with ten membrane-spanning domains [124,125]. It is highly expressed in neutrophils and, in lower levels, in endothelial cells and platelets (http://immprot.org, accessed on 7 January 2022). Given its homology with choline-transporter protein 1 (CTL1), SLC44A2 is suggested to have a transporter function, aiding the transport of choline, especially in the mitochondria [125,126,127,128]. A more recent study also suggests that SLC44A2 is important for the correct localisation of cellular adhesion proteins [129]. However, the cellular function of SLC44A2 is not well-defined. Its deficiency has been associated with hair cell loss, spiral ganglion degeneration and hearing loss in mice, [125,130] and with Meniere’s disease and TRALI in humans [124,131,132].

GWAS studies identified SLC44A2 as a susceptibility locus for VTE, despite the lack of any known links between SLC44A2 and thrombosis/coagulation [133,134,135]. Recent studies by Tilburg et al. also revealed that *Slc44a2^−/−^* mice exhibit normal haemostasis, but are protected against DVT [136,137]. They showed that *Slc44a2^−/−^* mice have venous thrombi with reduced weight and length in the IVC stenosis model, with a more profound effect being observed at earlier timepoints. A recent paper recapitulated some of these results in the stenosis thrombosis model of the IVC [128]. However, unlike previous studies where no effect of *Slc44a2* deficiency was reported on platelet function, the protection in thrombosis was partly attributed to defective platelet activation via lack of ADP/ATP production by mitochondria [128]. All these data clearly suggest the importance of SLC44A2 in DVT with a more prominent role in the initiation phase, rather than in the propagation phase of DVT [123,128,137].

Furthermore, GWAS studies identified a single nucleotide polymorphism (SNP) in *SLC44A2* (*rs2288904-G/A*), providing a 30–50% protection in VTE patients homozygous for the SNP [130]. This SNP is located in codon 461 of the gene, based on a substitution (G>A) that causes a missense mutation, R154Q in the first and longest extracellular loop of SLC44A2 [133,135,138,139,140].

We recently showed that neutrophils homozygous for the *rs2288904-A* SNP have an impaired ability to interact with VWF-‘primed’ platelets. These findings could, therefore, provide a causative link between the *rs2288904-G/A* polymorphism and the development of DVT [123]. Another recent study by Zirka et al. also shows that neutrophils homozygous for the SNP have a reduced ability to release NETs under venous shear rates, although under their experimental conditions they suggest that neutrophils can directly interact with VWF via this receptor which remains to be ascertained in vivo [141]. It is clear, however, from all these recent studies, that SLC44A2 is an important target to consider for future therapeutic strategies against DVT and that its molecular function warrants further investigation to shed more light upon the intricate mechanisms involved in the initiation of DVT.

At sites of disturbed venous flow, such as around venous valves, secreted VWF likely has an increased propensity to unravel and tangle. In tangling into cables, it in turn becomes more resistant to cleavage/removal by ADAMTS13. As a result, platelets could be captured via GPIbα and become ‘primed’ (Figure 3). VWF-‘primed’ platelets can, subsequently, recruit neutrophils via activated α_IIb_β_3_-SLC44A2 and trigger NETosis, providing a nidus for thrombus formation if these structures are not cleared. Neutrophils from individuals homozygotes for the *SLC44A2 rs2288904-A* polymorphism, (minor allele frequency of 22%), exhibited a diminished ability to bind VWF-‘primed’ platelets, and could therefore explain the reduction in the thrombus burden in these individuals. These findings provide mechanistic insights for understanding the link between this polymorphism and DVT protection, as well as offering important prospects for possible novel prophylactic/therapeutic strategies against DVT [123].

*GpIbα^−/−^* and *Vwf^−/−^* mice already exist, but *Vwf* deficiency also influences the release of both FVIII and P-selectin [142,143]. However, in both cases, VWF-platelet recruitment is completely abolished, making it impossible to study the A1-GPIbα downstream signalling events. The presence of the extracellular domain of GPIbα, as well as the VWF A1 domain are crucial for studying the role of this interaction. Moreover, it is important to preserve the filamin-binding site (a.a. 665–683 in mouse GPIbα) to prevent platelet cytoskeletal defects that could influence platelet phenotype. Indeed, complete deletion of GPIbα in mice, as well as GPIbα deficiency in humans, known as Bernard–Soulier syndrome, is associated with the presence of giant platelets (size increase from 1–2 μm to 4–10 μm) within the vasculature and abnormalities in proplatelet production [143,144,145]. Previous attempts to ablate the VWF A1-GPIbα mediated signalling without affecting platelet binding to VWF or disrupting the filamin-binding site were endeavoured. Kanaji et al. introduced a human *GpIba* transgene lacking the last six amino acids of the intracellular tail of GPIbα in *GpIbα^−/−^* mice [146]. However, although these mice displayed impaired thrombus formation in a ferric chloride carotid thrombosis model, [147] the authors reported no overt platelet defect or defective haemostasis [146].

In light of these studies, we generated a novel transgenic mouse, *GpIbα^Δ^^sig/^^Δ^^sig^*, by introducing an early stop codon after Pro694 by CRISPR-Cas9 technology, leading to the deletion of the last 24 amino acids of the intracellular tail of GPIbα (a.a. 695-718) [148]. This was hypothesised to completely (rather than partially) ablate the ability of 14-3-3 and phophoinositide 3-kinase (PI3K) to bind and mediate signalling events within platelets, while the filamin-binding site and ability of platelets to bind to the VWF A1 domain remain intact. Indeed, characterisation of these mice revealed that *GpIbα^Δ^^sig/^^Δ^^sig^* platelets bind normally to VWF under flow and have normal haemostasis [148]. Platelet responses were unaffected upon stimulation with thrombin or ADP, although platelets exhibited diminished GPVI-mediated signalling in response to CRP. As predicted, their platelets had an impaired ability to undergo VWF-GPIbα dependent signalling, as they formed fewer filipodia on VWF in the presence of botrocetin, and had reduced activation of α_IIb_β_3_ under flow [148]. These mice will allow formal investigation of the importance of VWF-GPIbα mediated signalling in vivo in thrombosis models.

As already mentioned, *Slc44a2^−/−^* mice have also been recently generated and studies suggests that these mice exhibit a normal haemostatic response, and, notably, a reduced risk of developing DVT [128,136,137]. However, SLC44A2 is also expressed, although at appreciably lower levels, in platelets and endothelial cells. Generating platelet/neutrophil/endothelial *Slc44a2* knockout mice to ascertain the role of SLC44A2 in DVT and other disease models would be an avenue of research to pursue in the future. Moreover, mice expressing the *rs2288904-A* variant of SLC44A2 could also be generated in order to establish the mechanistic link between this polymorphism and the protection against DVT.

## 6. Targeting Platelet–Neutrophil Interaction for Thrombosis Treatment

Currently, DVT thromboprophylaxis involves anticoagulant treatment primarily composed of low molecular weight heparin and/or vitamin K antagonists (e.g., warfarin). DOACs including apixaban, dabigatran or rivaroxaban are now increasingly given to VTE patients as an alternative due to their favourable pharmaco-kinetics, oral availability and the lack of need of monitoring [99,149,150,151,152]. However, whereas effective, anticoagulant therapy poses an increased risk of serious bleeding in treated individuals [99,150,151,152,153]. Dual antiplatelet therapies (usually aspirin in combination with a P2Y_12_ inhibitor) are widely used for the treatment of coronary artery disease but in some instances oral anticoagulants such as vitamin K antagonists can also be prescribed [154,155]. However, the increased benefit in reducing cardiovascular events with anti-platelet and anticoagulant combinations very often is accompanied by an increased bleeding risk [156]. Similarly, administering the current anti-platelet regimen in combination with the standard anticoagulation treatment in DVT/VTE has received some interest but the benefits are still unclear [157,158,159]. Interestingly, the SPATA-DVT study showed that the VTE risk is low in individuals with inherited platelet disorders that undergo surgery, suggesting there may be scope to target platelets for the treatment of VTE [160].

Consequently, the identification of novel strategies to target DVT or coronary arterial disease without affecting the bleeding risk is crucial. Due to its involvement, the intrinsic pathway has also been investigated as a potential target for VTE therapeutic agents [161], as this should be associated with a lower bleeding risk compared to the conventional anticoagulant therapies currently used. Both small molecule inhibitors and inhibitory antibodies have been assessed in animal models [162]. Dose-dependent effects on thrombus formation were observed with an anti-FXIa antibody in a vein thread-induced rabbit model, with no concurrent increases in bleeding [163]. Using a monoclonal antibody against FXI also proved to be effective in thrombus resolution within the early stages of DVT in mouse models [164]. Additionally, an inhibitory antibody against FXI, Abelacimab, as well as a selective inhibitor for FXI, Milvexian, were shown to be effective in preventing postoperative VTE in patients who had undergone knee arthroplasties [165,166]. Analogously, targeted depletion of FXII in mice with antisense oligonucleotides (ASO) resulted in reduced thrombus burden without alterations in haemostasis [167]. These preclinical studies have resulted in progression to clinical trials; both an ASO against FXI and an anti-FXIa antibody were shown to be superior to enoxaparin for VTE prevention in humans [168,169]. A small molecule inhibitor against FXIa is also undergoing clinical evaluation, with promising safety, pharmacokinetics and pharmacodynamics profile in a first-in-human study [170,171].

Inhibiting platelet–neutrophil interactions is a potential therapeutic avenue to explore for managing thrombosis without an increased risk of bleeding with for example targeting the P-selectin/PSGL-1 axis. Indeed, *Mac-1* deficient mice exhibited protection in models of thrombosis of the micro- and macro-circulation without altered haemostasis, with similar effects on thrombosis also observed with antibodies or glucosamine targeting the Mac-1/GPIbα interaction [172]. Moreover, Wong et al. showed that a PSGL-1 peptidomimetic reduced venous thrombus formation without influencing the normal haemostatic response [173]. Monoclonal antibodies or aptamers targeting P-selectin have also proven as efficacious as enoxaparin in reducing thrombosis in primate models of DVT [174,175,176].Very few clinical trials however, have been reported efficacious to date [177,178,179]. Infusion of Inclacumab—a recombinant monoclonal anti-P-selectin antibody—before PCI in non-STEMI patients reduced myocardial damage compared to placebo, holding promise that inhibiting P-selectin to PSGL-1 can be further explored therapeutically [178,179]. Indeed, a recent study showed that nanoparticles targeting PSGL-1 clustering on neutrophils showed efficacy in reducing both venous and arterial thrombosis in an antiphospholipid syndrome model, suggesting targeting the P-selectin/PSGL-1 axis may be beneficial in a broad range of thrombotic disorders [180].

Given the crucial role of NETs in both venous and arterial thrombosis, it is possible that an inhibitor of NET formation or enhancing NET dissolution would be beneficial. As already mentioned, recent studies in animal models propose the use of PAD4 inhibitors or DNases in the treatment of DVT or MI, to help prevent NET release or promote their dissolution, respectively [10,46,61,64,65,107,109,181,182,183]. Similarly, experimental mouse stroke models suggest that DNase alone or in combination with thrombolytic agents also have promising outcomes for stroke treatment [82,92,93]. More recently, asebogenin, a Syk phosphorylation inhibitor, was shown to reduce arterial thrombosis by decreasing platelet accumulation and fibrin formation, as well as venous thrombosis, by reducing neutrophil recruitment and subsequent NET formation [184]. However, all these approaches to target NET formation might equally affect physiological NETosis, which may impair the normal innate immune response. Clinical trials will ascertain if treated individuals might become more susceptible to infections.

Several GWAS studies have identified SLC44A2 as a susceptibility locus associated with VTE [133,135,139]. Individuals bearing the *rs2288904-A* polymorphism in SLC44A2 may be protected against DVT due to the impaired ability of their neutrophils to bind to activated α_IIb_β_3_ as shown by our data [123] and therefore to form NETs [123,141]. It can, therefore, be assumed that disrupting the interaction between VWF-‘primed’ platelets and neutrophils may contribute to DVT prophylaxis. Targeting the ‘primed’ platelet receptor, activated α_IIb_β_3_ with existing antiplatelet agents (e.g., Eptifibatide/Integrilin, Tirofiban) would affect platelet function and increase the risk in bleeding due to the implicit inability of platelets to aggregate. Intervening with the VWF-GPIbα binding (e.g., Caplacizumab) may be efficacious but would lead to a noticeable risk in bleeding [185]. Targeting the VWF-GPIbα signalling without disrupting the platelet capture may be achievable with small permeable inhibitors if selective targeting within the platelet could be achieved [186,187]. In contrast, SLC44A2 has little influence on the haemostatic function [136,137]. Thus, targeting SLC44A2 would not be predicted to be associated with a bleeding risk in treated individuals and therefore represents an attractive target for thromboprophylaxis potentially as an adjunctive approach [123].

## 7. Platelet and Neutrophil Interactions in COVID-19

As of November 2022, there have been more than 600 million confirmed cases of coronavirus and over 6.5 million deaths worldwide. The coronavirus disease (COVID-19) is caused by severe acute respiratory syndrome coronavirus 2 (SARS-CoV-2), which initially manifests as flu-like symptoms. However, in severe cases, there is excess generation of pro-inflammatory cytokines (cytokine storm) leading to acute respiratory disease syndrome and in those worst affected, multi-organ failure [188]. Alongside characteristic symptoms affecting the respiratory system, venous and arterial thrombotic complications have been implicated in COVID-19 pathology and mortality. These include platelet and fibrin microthrombi in the lung, heart, liver and kidney microvasculature, alongside high rates of DVT and stroke [189,190]. Additionally, thrombocytopenia has been associated with worse outcomes in COVID-19 patients [191]. As highlighted throughout this review, the interplay between thrombosis and inflammation is becoming ever-more apparent. Hence, activation of platelets and leukocytes, particularly neutrophils, via inflammatory mediators has become a key line of investigation in understanding thrombotic mechanisms in COVID-19.

Initial studies highlighted the link between SARS-CoV-2 infection and platelet hyperactivity [192,193,194]. Platelet activation in COVID-19 has been identified through plasma markers of alpha and dense granule release, activation of α_IIb_β_3_ and presence of platelet-derived extracellular vesicles [192,193,195,196,197]. Furthermore, platelets isolated from COVID-19 patients exhibited increased activation and aggregation in response to agonists, as well as spreading on collagen/fibrinogen [192,193,195,198,199]. In line with platelet hyperactivity, the propensity of platelets to release of inflammatory molecules may also be increased in COVID-19. Levels of numerous inflammatory cytokines, including those important for recruitment and activation of neutrophils (e.g., CD40L, RANTES), were increased in plasma and reduced in platelet lysates from patients, in response to low dose thrombin [192,200]. This highlights the increased reactivity of platelets and their possible contribution to the inflammatory milieu observed in COVID-19.

As neutrophils are involved in the first line of host defence against pathogens and their association with lung injury, alterations to neutrophil function and activation of NET formation were an immediate focus for research in COVID-19 pathogenesis. COVID-19 patients exhibited elevated levels of leukocytes, particularly neutrophils, while neutrophils/NETs were identified within thrombi throughout the pulmonary, hepatic and renal vasculature, with the highest burden observed in the lungs [201,202,203,204,205]. Consistent with this, plasma extracellular DNA, MPO-DNA, DNA-NE complexes and CitH3, all markers of NETs, were elevated in COVID-19 patients [200,206,207]. Finally, anti-NET antibodies which may impair NET clearance have been identified in COVID-19 patients and were correlated with the disease severity [208]. All together these studies highlight the contribution of NET-releasing neutrophils to the COVID-19 pathology.

Due to the importance of platelet–neutrophil interactions in NET formation, it is unsurprising that several studies reported increased numbers of platelet–neutrophil aggregates in COVID-19 patients [193,200,201,204,209]. In one study, these platelet–neutrophil aggregates correlated with disease severity [193]. Studies on influenza and HIV have previously delineated signalling pathways starting with internalisation of virions by platelets to platelet–neutrophil interactions via P-selectin and CD40L surface expression [210]. Indirect stimulation of NETosis via this mechanism may also occur through platelet TLR7 activation by virion uptake, resulting in complement component 3 release and stimulation of NET formation [210]. However, it is still debated whether platelets internalise SARS-CoV-2 [192,193,211,212]. The signalling mechanisms between platelets and neutrophils to induce NET formation within COVID-19 have not been entirely elucidated. Thrombin-PAR1 and complement signalling have been implicated, with inhibition of thrombin or blockade of C5aR1 attenuating platelet-mediated NETosis [213]. A recent elegant study also revealed a novel mechanism by which S100A8/A9 (released from myeloid cells including neutrophils) can bind GPIbα, drive the formation of procoagulant platelets and facilitate fibrin formation. As previously shown in other studies, [205,214,215,216] elevated plasma levels of S100A8/A9 correlated with the severity of the disease [204]. All together in COVID-19 patients, platelet and neutrophil hyperreactivity and crosstalk participate in excessive immunothrombosis resulting in coagulopathy, thrombosis and respiratory failure.

## 8. Conclusions

Although platelets have classically been classified as key players in haemostasis and neutrophils as vital cells of our innate immune system that fight infections, it is clear that platelets carry important immune functions and neutrophils via NET formation facilitate thrombus development. There is compelling evidence for an important role for platelet–neutrophil interactions in both arterial and venous thrombosis. The influence of their activation state on each other and how it feeds the vicious circle of inflammation–coagulability–thrombosis is central to the aetiology of many cardiovascular diseases. Understanding initial events leading to platelet–neutrophil interactions and subsequent NET formation in vivo in various disease settings will be important to develop additional therapies to prevent thrombosis without increasing the bleeding risk or compromising the immune responses of patients. The α_IIb_β_3_-SLC44A2 axis in the context of DVT represents a potential candidate to develop such a therapy in the future, although we are just beginning to understand the molecular mechanisms downstream of this interaction. Can the concept of platelet priming be mediated by factors other than VWF? What are the signalling pathways involved downstream of the binding of SLC44A2 to α_IIb_β_3_ that triggers NET formation? Is α_IIb_β_3_ the only ligand for SLC44A2? Identifying the precise binding site(s) of SLC44A2 on α_IIb_β_3_ will also greatly facilitate the design of SLC44A2 inhibitors. Ultimately, identifying additional molecular targets of immunothrombosis is warranted to provide additional therapeutic approaches to complement anti-platelet, anti-coagulant and anti-inflammatory treatments to fight thrombosis including in the context of infections such as COVID-19.

## Figures and Tables

**Figure 1 ijms-24-01266-f001:**
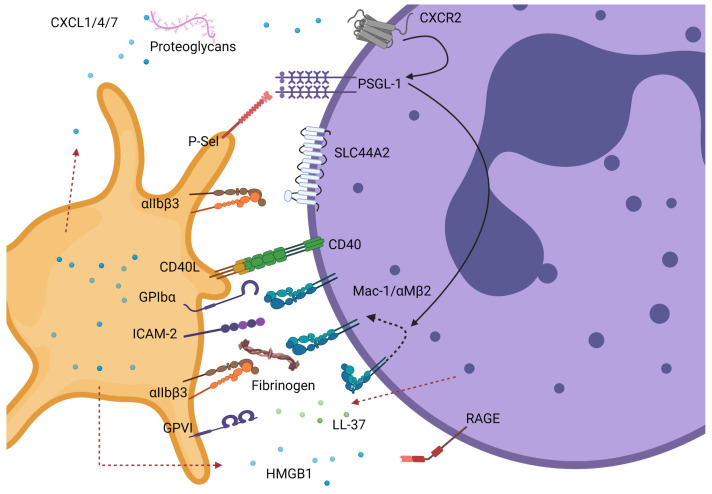
Known interactions between platelets and neutrophils. These can be mediated by direct contact between platelet and neutrophil receptors or indirectly via secreted ligands or molecules present in plasma. Most of these interactions require either the platelet and/or the neutrophil to be activated. Neutrophils can interact via PSGL-1 or CD40 with P-selectin and CD40L, respectively, on activated platelets. PSGL-1 binding to P-selectin triggers downstream signalling in neutrophils (black solid lines) leading to Mac-1 integrin conformational change from a closed to open conformation (dotted black line). These events can be enhanced by cooperative signals from CXCL1/4/7 immobilised by proteoglycans binding to CXCR2 or Mac-1. Activated Mac-1 can further promote binding to platelets directly via GPIbα or ICAM-2 or indirectly via fibrinogen-bound α_IIb_β_3_. Additional depicted interactions are granule-release HMGB1 from platelets with neutrophil RAGE receptor promoting further neutrophil activation and LL-37 released from neutrophils (dotted red line) activating platelets via GPVI. Lastly, direct binding of activated α_IIb_β_3_ to neutrophil receptor SLC44A2 can trigger neutrophil activation and NET formation. GPIbα- glycoprotein Ibα; GPVI- glycoprotein VI; ICAM-2- intracellular adhesion molecule 2; CD40L- CD40 Ligand; P-sel- P-selectin (CD62P); CXCL- C-X-C Motif Chemokine Ligand; HMGB1- High mobility group box 1 protein; PSGL-1- P-selectin glycoprotein ligand; SLC44A2- solute carrier family 44 member 2 or CTL-2; MAC-1- macrophage 1 antigen or CD11bCD18 or α_M_β_2_; LL-37—human cathelicidin antimicrobial peptide; RAGE- receptor of advanced glycation end products. Created by Biorender.com.

**Figure 2 ijms-24-01266-f002:**
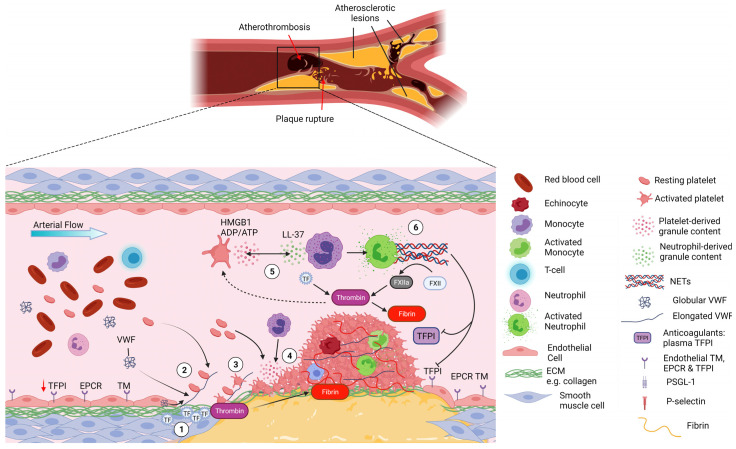
Platelet–neutrophil interactions in atherothrombosis. Atherothrombosis is characterised by atherosclerotic plaque rupture or erosion, exposing collagen-rich plaque material to the blood and causing subsequent thrombus formation. **1-** Upon plaque rupture the subendothelial matrix becomes exposed, allowing VWF to bind collagen and exposing TF, initiating the extrinsic coagulation pathway. **2-** Upon the shear forces of the blood, VWF unravels and binds platelets via GPIbα. The fast on and off rates of the VWF-GPIbα interaction allow the deceleration of platelets and their binding to collagen receptors. **3-** Platelets become activated and start forming aggregates while releasing their granule content. **4-** This promotes the recruitment of additional platelets but also neutrophils to the site of vascular injury via HMGB1 binding to neutrophil RAGE receptor. **5-** Once recruited, neutrophils become activated and release their granule content which further activates platelets (e.g., LL-37 activating GPVI platelet receptor). Blood born TF from monocyte microvesicles further augments the extrinsic coagulation pathway leading to thrombin release in the milieu. **6-** Progressively, the thrombus grows, mostly composed of platelets and fibrin that consolidate the thrombus but with the presence of neutrophils. Platelet-induced activation of neutrophils triggers the formation of NETs that further augment the coagulation system via activation of the contact pathway and inhibiting anticoagulant TFPI. Ultimately, the overwhelming activation of the coagulation pathways and platelets lead to excessive thrombus formation and arterial occlusion in myocardial infraction and stroke. Created by Biorender.com.

**Figure 3 ijms-24-01266-f003:**
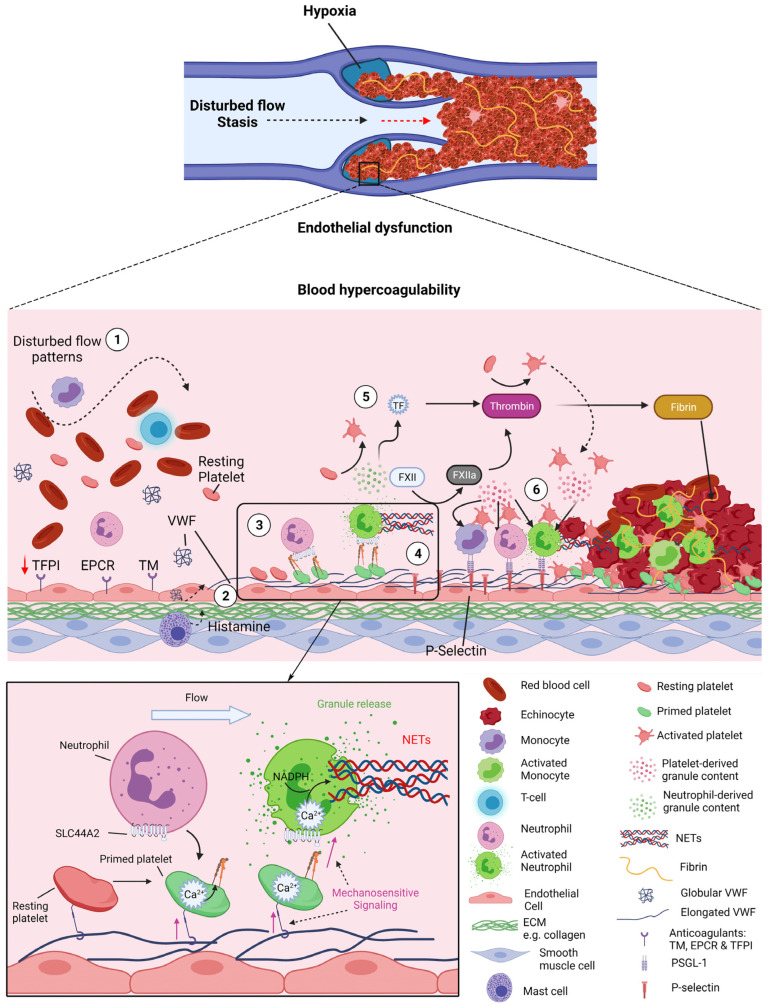
Platelet–neutrophil crosstalk in venous thrombosis. Venous thrombosis initiates in the pockets of venous valves where disturbed flow creates a prothrombotic environment: hypoxia, endothelial dysfunction and blood hypercoagulability. **1-** These flow patterns/stasis induces downregulation of antithrombotic factors of the endothelium thrombomodulin (TM), endothelial protein C receptor (EPCR) and tissue factor pathway inhibitor (TFPI) and concomitant upregulation of procoagulant proteins von Willebrand Factor (VWF), P-selectin and other cell adhesion molecules. Mast cells are also thought to be involved in this process by releasing histamine. **2-** Under these flow disturbance and pro-coagulant conditions, VWF can become tangled and unravel, exposing its A1 domain enabling platelet binding via GPIbα. **3-** The VWF-GPIbα interaction under flow induces the mechano-unfolding of the mechanosensitive domain of GPIbα leading to signalling events, Ca^2+^ release from intracellular stores and activation of α_IIb_β_3_. **4-** Primed platelets via activated α_IIb_β_3_ can bind SLC44A2 on neutrophils under flow. Shear forces in the neutrophil induce calcium- and NADPH-mediated NET formation. **5-** Activated neutrophils can directly activate platelets through release of granules (e.g., LL-37) or indirectly by generating thrombin via NETs. NETs can directly activate the intrinsic coagulation pathway by binding to FXII and inhibit the anticoagulant protein TFPI, augmenting thrombin generation. Granule content from both platelets and neutrophils further stimulates the endothelium leading to increased expression of P-selectin and other adhesion molecules. **6-** Monocytes and neutrophils are recruited to the endothelium via PSGL-1 binding to endothelial P-selectin. Activated platelets are able to bind activated neutrophils and monocytes and further stimulate them by release of their content (e.g., HMGB1, cytokines). NETs promote thrombus development by binding red blood cells, VWF and platelets, activating of both intrinsic and extrinsic coagulation pathways, and also facilitate thrombus stability by providing fibrinolytic resistance. Ultimately venous thrombi rich in red blood cells and fibrin obstruct the valve and upstream vein, causing DVT. Created by Biorender.com.

## Data Availability

Not applicable.

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
