# Peer review of "Platelet–Neutrophil Crosstalk in Thrombosis"

_ijms, 2023, doi:10.3390/ijms24021266_

Round 1

Reviewer 1 Report

The authors carried out a very detailed review, extremely well documented and including the most recent data on how platelets and neutrophils interacts and the consequences of these interactions in the pathophysiology of thrombosis. They made the distinction between arterial and venous thrombosis and they listed a certain number of targets potentially of interest for the development of antithrombotic drugs with a low risk of bleeding. The risk inherent to this type of review dealing with such a complex subject with such a large and sometimes contradictory bibliography is not to lose the reader between what is formally established and what is of the order of the prospective. 

Thus, it is sometimes difficult to distinguish the hierarchy between the different actors and the different pathways according to the site and the etiology of thrombosis. 

This reservation relates more particularly to the part of the manuscript devoted to arterial thrombosis.

-      If it is true that neutrophils participate in the development of the atherosclerotic plaque even if they are not the main players, but atherogenesis is not the subject of the review. The corresponding paragraph on page 5 from line 172 could therefore be lightened. Concerning thrombosis on the ruptured atherosclerotic plaques, if neutrophils may have a role, they are not the starter. All the studies carried out by perfusing blood on extracts of atherosclerotic plaques have underlined that the two key events for the formation of the thrombus are GPVI-mediated platelet activation by the highly abundant and prothrombotic collagen of the plaque on the one hand and the tissue factor triggered thrombin generation on the other hand. These assays don’t so far provide evidence of a significative contribution of neutrophils in thrombus formation growth and stability. This does not exclude that neutrophils can be recruited by the MI thrombus in vivo, in particular by activated platelets and fibrin. However, the presence of cathelicidin and NETs in MI thrombi and of circulating S100A8/9 may be more markers of neutrophil recruitment than the evidence of a major pathogenic role of neutrophils in MI. Moreover, the mechanisms of neutrophil recruitment and their potential effects described in this section of the manuscript are not specific to arterial thrombosis and seem to be of greater importance in venous thrombosis, which is more clearly a matter of thrombinflammation than arterial thrombosis.

The part on ischemic stroke is very important but it is unfortunate that the authors did not differentiate the role of platelet/neutrophil interactions at the level of the culprit thrombus on the one hand and at the level of the downstream microcirculation on the other. As correctly reported by the authors, histological analysis of culprit thrombi obtained by thrombectomy have underlined the importance of NETs in the structure of the thrombus and the resistance to tPA. Additionaly, at the level of the microcirculation, intravital microscopy in rats and mice have shown the development of thromboinflamation very early after the onset of ischemia leading to the occusion of venules. Even when recanalization procedures are successful,  lack of reperfusion of cerebral microcirculation (no reflow) is associated with poor prognosis and increased risk of intracranial bleeding. In animal models of ischemic stroke, both GPIb and GPVI have been reported to be critical in cerebral thromboinflammation. The part of the manuscript dedicated to ischemic stroke should thus be modified to introduce these differents aspects of platelet interaction with neutrophils with some representative references such as those of Desilles’ or Stoll’s teams.( J Am Heart Assoc . 2018;7:e007804. DOI: 10. 1161/JAHA.117.007804.  Nature reviews Neurology 2019 ;  15 :473.  https://doi.org/10.1038/s41582-019-0221-1)

One suggestion of the reviewer would then to  first describe in a general manner the interactions of platelet with neutrophil interactions with the description of molecular partners and consequences on both platelets and neutrophils for the formation, growth and stability of the thrombus. Then in a second time the specificities of these processes at the arterial, venous and microvascular level could be precised. It would also be wise to mention, if known, the effect of shear forces on platelet interactions with neutrophiles and their conséquence. As an example do shear forces influence NETs production?

Interestingly, the authors propose a list of potential targets to pharmacologicaly regulate platelet interaction with neutrophils and their consequences. The reviewer understands that the list cannot be exhaustive. Nevertheless, it would be wise to add platelet GPVI to this list. Indeed GPVI is receiving increasing attention as a potential target for effective and safe antiplatelet therapy and studies in animal models of stroke have shown that GPVI contributes to thromboinflammation. In addition, GPVI antagonists are tested in clinical trials in stroke with encouraging results. An other suggestion is to present the different targets and their antagonists as a table.

The part of the manuscript dédicated to COVID-19 is not appropriate because many reviews have aleady described thrombinflammation in this pathology. In addition, none of the clinical trials that have tested antiplatelet therapies in patients hospitalized or not for moderate to severe COVID-19 have shown any efficacy on the evolution of the pathology. (see Zong X et al. Front. Med. 9:965790. doi: 10.3389/fmed.2022.965790) In addition, this part relates to the roles of platelets in host defense against infections, which is beyond the scope of this review. It is preferable that the review remains focused on platelets/neutropils in sterile thrombosis.

Reviewer 2 Report

In this manuscript, the authors presented comprehensive review on how platelets and neutrophils interact with each other and how their crosstalk contributes to both arterial and venous thrombosis and in COVID-19. Targeting this interaction may provide a novel approach for treating thrombosis without compromising normal hemostasis. This work is very nicely written and is very pleasant to read. The graphic illustrations are also very nice.

Minor comments:

In section 5 “Targeting platelet-neutrophil interaction for thrombosis treatment” the authors first presented data suggesting the potential benefit of targeting the intrinsic pathway, then discussed the downside of traditional anticoagulation strategies, followed by 3 paragraphs discussing the potential of targeting platelet-neutrophil interaction. Maybe the order of these paragraphs should be rearranged: fist discuss the downside of traditional anticoagulation strategies, then discuss the novel strategies.

Page 14, line 594. It is difficult to conclude a direct contribution of increased platelet reactivity to the inflammatory milieu observed in COVID-19.

Some information of some references are incomplete, e.g. ref#91, #145, #146. Please double check.
